# Clinical and Molecular Features of Malignant Pleural Effusion in Non-Small Cell Lung Cancer (NSCLC) of a Caucasian Population

**DOI:** 10.3390/medicina60111804

**Published:** 2024-11-03

**Authors:** Irene Lojo-Rodríguez, Maribel Botana-Rial, Almudena González-Montaos, Virginia Leiro-Fernández, Ana González-Piñeiro, Cristina Ramos-Hernández, Alberto Fernández-Villar

**Affiliations:** 1Pulmonary Department, Alvaro Cunqueiro Hospital, PneumoVigoI+i Research Group, Sanitary Research Institute Galicia Sur (IISGS), 36312 Vigo, Spain; irene.lojo.rodriguez@sergas.es (I.L.-R.); almudena.gonzalez.montaos@sergas.es (A.G.-M.); cristina.ramos.hernandez@sergas.es (C.R.-H.); 2Pulmonary Department, Alvaro Cunqueiro Hospital, PneumoVigoI+i Research Group, Sanitary Research Institute Galicia Sur (IISGS), CIBERES-ISCIII, 36312 Vigo, Spain; virginia.leiro.fernandez@sergas.es (V.L.-F.); alberto.fernandez.villar@sergas.es (A.F.-V.); 3Pathology Department, Alvaro Cunqueiro Hospital, Spain PneumoVigoI+i Research Group, Sanitary Research Institute Galicia Sur (IISGS), 36312 Vigo, Spain; ana.gonzalez.pineiro@sergas.es

**Keywords:** malignant pleural effusion, diagnosis, biomarker, lung cancer, survival, targeted therapy

## Abstract

***Background and Objectives:*** The diversity of patients with malignant pleural effusion (MPE) due to non-small cell lung cancer (NSCLC) as well as the variability in mutations makes it essential to improve molecular characterization. ***Objective:*** Describe clinical, pathological, and molecular characteristics MPE in a Caucasian population. ***Materials and* *Methods:*** Retrospective study of patients with NSCLC diagnosis who had undergone a molecular study from 1 January 2018–31 December 2022. Univariate analysis was performed to compare patient characteristics between the group with and without MPE and molecular biomarkers. ***Results:*** A total of 400 patients were included; 53% presented any biomarker and 29% had MPE.PDL1, which was the most frequent. EGFR mutation was associated with women (OR:3.873) and lack of smoking (OR:5.105), but not with MPE. Patients with pleural effusion were older and had lower ECOG. There was no significant difference in the presence of any biomarker. We also did not find an association between the presence of specific mutations and MPE (22.4% vs. 18%, *p* = 0.2), or PDL1 expression (31.9% vs. 35.9%, *p* = 0.3). Being younger constituted a protective factor for the presence of MPE (OR:0.962; 95% CI 0.939–0.985, *p* = 0.002), as well as ECOG ≤ 1 (OR:0.539; 95% CI 0.322–0.902, *p* = 0.01). ***Conclusions:*** This is the first study that describes the clinical, pathological, and molecular characteristics of MPE patients due to NSCLC in a Caucasian population. Although overall we did not find significant differences in the molecular profile between patients with MPE and without effusion, EGFR mutation was associated with a tendency towards pleural progression.

## 1. Introduction

Malignant pleural effusion (MPE) is common in patients with cancer. Most MPE are secondary to metastasis to the pleura, most of them from lung cancer [1,2,3]. The presence of MPE indicates advanced disease and poor survival, and in the patients with lung cancer it upstages the cancer to stage IV [4].

Lung cancer (LC) is the leading cause of cancer death in the United States and around the world. LC has been the most common cancer worldwide since the 1980s, both in terms of incidence and mortality, affecting 13% or 1.6 million of total cancer cases. It can be separated into two main forms: small cell lung cancer (SCLC), a highly malignant tumor derived from cells exhibiting neuroendocrine characteristics, which accounts for 15% of lung cancer cases; and non-small cell lung cancer (NSCLC), which accounts for the remaining 85% of cases and is the one we are referring to in this article [4].

However, for patients with NSCLC, the last decade has been characterized by the critical progress of treatment options, which has contributed to a substantially improved survival [5]. MPE can occur in 15% of patients with LC and it is a severe condition of advanced tumors without effective therapy. Management should be prompt and care plans should be individualized and involve a multidisciplinary team of healthcare professional [3,4].

Management should be timely, and care plans must be personalized, involving a multidisciplinary team of healthcare professionals. Precision oncology involves strategies aimed at tailoring cancer treatments based on tumor biology. A notable group of patients with non-small cell lung cancer (NSCLC) has actionable genomic alterations that can benefit from targeted therapies. In LC, mutations in the epidermal growth factor receptor (EGFR) and rearrangements of the anaplastic lymphoma kinase (ALK) are well-established oncogenic drivers. The therapeutic application of tyrosine kinase inhibitors (TKIs) has shown better outcomes, compared to chemotherapy, for these mutations [6,7]. Other druggable targets have also been identified, with effective inhibitors developed and commercialized, leading to a significant shift in the treatment of NSCLC [8]. However, the presence of malignant pleural effusion (MPE) is linked to a poor prognosis in patients with EGFR-mutated NSCLC undergoing first-line treatment with TKIs [9]. The management of MPE in NSCLC patients with EGFR mutations may differ from those without these mutations. In patients with EGFR mutations treated with TKIs, early talc pleurodesis may not provide additional benefits in preventing pleural fluid buildup [10].

More recently, the development of immune checkpoint inhibitors has completely revolutionized the management of NSCLC [11,12].

Particularly, the creation of targeted antibodies against the programmed death receptor (PD-1), programmed death-ligand 1 (PD-L1), and the cytotoxic T-lymphocyte-associated protein 4 receptor has significantly improved treatment strategies for metastatic NSCLC, whether used as first-line or second-line therapies, resulting in the unprecedented extended survival of some patients [12]. Particularly talking about MPE by NSCLC, there is not much evidence, though a positive rate of PD-L1 expression in around 33% of cases has been described [13].

The diversity in the evolution and prognosis of patients with MPE due to NSCLC, as well as the variability in the frequency of mutations, makes it essential to improve the clinical and molecular characterization of these patients [6]. Most evaluations of these molecular biomarkers are carried out in small and/or heterogeneous populations and, mostly in Asian populations, which may not necessarily yield results similar to ours.

The objective of this study is to describe the clinical, pathological, and molecular characteristics of patients with MPE due to NSCLC in a Caucasian population; see what factors can be associated with the presence of biomarkers; and compare the results with patients with NSCLC without MPE.

## 2. Materials and Methods

### 2.1. Study Population

This is a retrospective study that included all patients with a diagnosis of NSCLC who had been the subject of a molecular study from 1 January 2018 to 31 December 2022. The following inclusion criteria were considered:Patients with diagnosis of advanced or locally advanced NSCLC at the time of inclusion in the study;Patients ≥ 18 years old;Hispanic patients;Patients with advanced or locally advanced NSCLC that underwent molecular testing.

Exclusion criteria were as follows:(a)Patients without a confirmed diagnosis of NSCLC;(b)Histological tumors other than NSCLC.

The data source was the Pathological Anatomy registry of the Álvaro Cunqueiro Hospital, a tertiary hospital with a health area of more than 600,000 patients.

A patient was defined as having MPE secondary to NSCLC when there was a cytohistological confirmation of neoplasia in the pleural fluid samples and/or pleural biopsy or in those patients with paramalignant PE. Paramalignant PE can occur due to bronchial obstruction by a tumor, leading to the subsequent atelectasis and development of ipsilateral pleural effusion. It can also result from pulmonary embolism, mediastinal lymphatic obstruction, thoracic duct obstruction (chylothorax), superior vena cava obstruction, malignant pericardial effusion and cardiac tamponade, hypoalbuminemia, and the side effects of chemotherapy or radiotherapy. Other causes of effusion (parapneumonic PE or congestive heart failure for example) were ruled out.

### 2.2. Study Variables

We analyzed the clinical-pathological characteristics (age, sex, stage IIIB vs. IV, histology, smoking status, symptoms at diagnosis, and other diseases) included in the index Charlson. Performance status (PS) was categorized from 0 to 4, according to the scale of the Eastern Cooperative Oncology Group (ECOG). The type of treatment administered and the chemotherapy lines were also analyzed. Treatment indication was based on the decision of the multidisciplinary committees and on the clinical practice guidelines in force at the time of the study. (https://www.nccn.org/professionals/physician_gls/pdf/nscl.pdf; accessed data: 1 April 2023).

Regarding patients with MPE, the pleural therapeutic option that was made to control MPE and the specific prognostic variables, described in the LENT scale [14], were recorded. The radiological variables from the computed tomography (CT) were described, including the amount of MPE and the presence of pleural thickening and/or nodules. Whether PE occurred at diagnosis or at progression, and the time until progression were recorded.

Finally, survival was recorded.

### 2.3. Molecular Analysis

Oncological treatment, selection of tested genes and their techniques were based on the clinical practice guidelines in force at the time of the study (NCCN guideline NSCLC, www.nccn.org). Molecular testing, the epidermal growth factor receptor (EGFR), anaplastic lymphoma kinase (ALK), c-ros oncogene 1 (ROS1), and programmed death-ligand 1 (PD-L1), were determined in tumor samples in all patients included in the study.

Molecular tests were performed on formalin-fixed paraffin-embedded blocks using a wide range of methods. Each molecular mutation had its own protocol as follows:-Sanger sequencing was used for detection of EGFR mutations within exons 18–21, using 5 tissue sections at 10 microns, DNA extraction following the commercial method and real-time PCR CobasR with a cut for HE, subsequent verification of the percentage of tumor cells in the sample;-Reflex fluorescence in situ hybridization testing was performed to detect ALK gene rearrangement. Using a 4-micron tissue section for ALK IHC with the VENTANA Anti-ALK clone (D5F3) and using a 3-micron tissue section for FISH with the Zytolight SPEC ALK Dual Color Break Apart probe from Zytovision, in those cases with an equivocal IHC result;-ROS1 rearrangements were detected by real-time polymerase chain reaction (RT-PCR). Detection of ROS-1 gene rearrangement using a 3-micron tissue section for immunohistochemistry using Ventana clone SP384 and a 3-micron section for FISH with the Zytolight SPEC ROS1 Dual Color Break Apart probe from Zytovision were carried out in case of doubtful or positive results with immunohistochemistry;-PD-L1 expression was assessed during screening by means of the PD-L1 IHC 22C3 pharmDx assay in formalin-fixed tumor samples. PD-L1 expression was considered positive when more than 1% of cells expressed it.

### 2.4. Ethical Aspects

The study was conducted following the Helsinki Declaration guidelines. The data were pseudonymized for biomedical research purposes. The study has been approved by the Provincial Research Ethics Committee of Galicia (Pontevedra-Vigo-Ourense code: 2022/18). The need to obtain informed consent was waived by the Ethics Committee of Galicia (Pontevedra-Vigo-Ourense code: 2022/18).

### 2.5. Statistical Analyses

Descriptive analyses were performed to describe the characteristics of the patient sample (mean, SD, percentages, and frequencies). All categorical variables were analyzed using Chi-squared tests, and the continuous variables were analyzed with Student’s *t*-tests.

Univariate analysis was performed to compare patient characteristics between the group with MPE and without MPE and the groups with and without molecular biomarkers. To perform this test, continuous variables had to be dichotomized. We used multivariate logistic regression models to determine the predictors of results. A *p* value of <0.05 was considered statistically significant. All analyses were performed using Statistical Package for the Social Sciences (SPSS) software (version 20.0 for Windows; SPSS Inc., Chicago, IL, USA). With a α-risk of 0.05, and a β-risk of 0.2 (80% power), we estimated that the study needs to recruit a total of 80 patients with MPE.

### 2.6. Data Availability

The datasets generated and/or analyzed during the current study are not publicly available due to the ongoing investigation being part of a doctoral thesis but are available from the corresponding author on reasonable request.

## 3. Results

During the study period, 400 patients with a diagnosis of advanced or locally advanced NSCLC and with molecular determinations performed were included.

The demographic and clinical characteristics of the entire sample are represented in Table 1. Regarding molecular characteristics, almost 53% of the cases presented any biomarker, the most frequent being the expression of PDL1 in 35% of the cases and the EGFR gene mutation in 14% (Figure 1).

Oncological treatment was carried out in 65.3% (261/400) of patients, mainly through immunotherapy or TKI (47.1%—123/261). The indication for oncological treatment was greater in patients with a positive biomarker (59.8%—156/261 vs. 40.2%—105/261, *p* = 0.0001).

The EGFR gene mutation was the most recurrent in patients with advanced NSCLC. The characteristics of patients with the EGFR gene mutation expression are shown in Table 2. Tumor recurrence, such as pleural involvement, was more prevalent in patients with the mutated EGFR gene.

However, in the multivariate analysis, the presence of the EGFR gene mutation was independently associated with female sex (OR: 3.873; 95% CI 1.084–13.836, *p* = 0.04) and the absence of smoking (OR: 5.105; 95% CI 1.363–19.120, *p* = 0.02), but not with the presence of pleural effusion.

Of the 400 cases included in the analysis, 29% (116/400) had MPE. At the time of diagnosis of NSCLC, 80.2% (93/116) presented it, while 19.8% (23/116) presented it as their disease progressed, with a mean onset time of 11.7 ± 11.1 months.

Table 3 describes the characteristics of patients with and without MPE. Patients with pleural effusion were older, smoked less, and had lower ECOG. The presence of nodules or pleural thickening was also significantly associated with the presence of pleural effusion. However, there were no statistically significant differences in the presence of any biomarker overall, including PDL1 expression, in cases with or without pleural effusion. When the analysis was performed based on the type of molecular biomarker, we also found no association between the presence of specific mutations (EGFR, ALK, and ROS1) and MPE (22.4% vs. 18%, *p* = 0.2). Additionally, no association was found in the expression of PDL1 (31.9% vs. 35.9%, *p* = 0.3), nor in the specific case of the most frequent alteration, the mutation of the EGFR gene (18.1% vs. 13.4%, *p* = 0.3).

In a multivariate analysis, it was found that being younger constituted a protective factor for the expression of MPE (OR: 0.962; 95% CI 0.939–0.985, *p* = 0.002), in addition to a lower ECOG (ECOG ≤ 1) (OR: 0.539; 95% CI 0.322–0.902, *p* = 0.01). Likewise, pleural thickening was also independently associated with the risk of suffering from MPE (OR: 14.994; 95% CI 4.731–47.521, *p* = 0.001).

A therapeutic intervention on the effusion was performed in 44.8% (52/116) of the patients, with indwelled pleural drainage being the choice in 40.4% (21/52) of the patients. In the group of patients who were definitively treated for DPM, the oncological treatment offered was similar to the group without treatment for MPE (44.9%—31/69 vs. 55.1% 38/49, *p* = 0.6).

The mean overall survival was 11.6 months (p25–2, p75–16), being significantly lower in patients with pleural effusion compared to patients without MPE (12.5 ± 19.0 vs. 9.5 ± 11.9, *p* = 0.05). Offering a definitive solution for MPE did not result in changes in survival (8.4 ± 8.8 vs. 10.4 ± 13.9 months, *p* = 0.4).

## 4. Discussion

To our knowledge, this is the first study that describes the clinical, pathological, and molecular characteristics of patients with MPE due to NSCLC in a Caucasian population. Although overall we did not find significant differences in the molecular profile between patients with MPE and those without effusion, the presence of the EGFR gene mutation was one of the most frequent biomarkers found and was associated with a tendency towards tumor progression in the form of pleural metastases.

MPE is one of the main causes of exudate that we see in clinical practice, with NSCLC being one of the most frequent causes [1,2,3]. The pathogenesis of MPE is very complex and there may be tumor infiltration via hematogenous, direct, or even lymph node blockade. It is not clear why some patients develop MPE and others do not, but it is known that there is a higher incidence in patients with larger tumors, mediastinal involvement, or in the adenocarcinoma subtype [1,2,3,4]. Patients with a MPE are grouped as stage IV, M1a category, in the TNM classification of NSCLC, as the overall survival (OS) of this patient population is less than one year. In our study we included patients with diagnosis of advanced or locally advanced NSCLC at time of inclusion in the study. For this reason, we included patients with initial pleural effusion and patients who developed a MPE after. In our series it is reflected that MPE was more recurrent in older patients and with worse ECOG. As we already mentioned, we did not find differences in the molecular characteristics between patients with MPE and those without pleural effusion.

Yong Zou et al. [15] performed a retrospective study in a single center with the objective of comparing the clinical features of patients with lung adenocarcinoma with and without EGFR mutations. Authors described that in patients with stage IV disease, the frequency of EGFR mutations was higher in those with MPE than in those without MPE. EGFR mutations were independently associated with female sex, no history of smoking and the presence of MPE. The conclusions were that there was a positive association between EGFR mutation and the presence of MPE. EGFR mutations may play an important role in the formation of MPE [9]. We found that the presence of mutated EGFR was not significantly more frequent in patients with MPE. We observed that it was associated with a higher frequency of pleural recurrence. This shows the importance of studying different populations. As in our study, the determination of the EGFR gene mutation was performed in the primary tumor [15].

Wang S. [16] compared EGFR mutations detected in MPE, plasma, and tissue in patients with lung adenocarcinoma. MPE was a reliable surrogate for tumor tissue in identifying EGFR mutations. MPE could offer a reference for the EGFR mutation to inform the EGFR-TKI treatment decision for advanced lung adenocarcinoma patients, even when tissue and plasma were available [16].

At present, determining a treatment subsidiary mutation has become essential. Han et al. [14] compared DPM samples and primary tumor samples in patients with lung adenocarcinoma. The presence of EGFR gene mutations in patients with MPE was significantly higher, indicating that these mutations could facilitate cancer migration to the pleural cavity [15].

MPE is more often associated with the mutated EGFR gene and the ALK fusion gene in Asian populations [17]. In our study the EGFR gene mutation rate was 14.8%, much lower than in other studies in the Asian population [15,17]. This reflects the importance of conducting this type of study in different populations, since the findings may not be generalized to other racial or ethnic groups [18,19,20].

A limitation of our study is that all molecular determinations were not performed on pleural fluid cell block or pleural biopsy samples. Scientific evidence has shown that the study of these biomarkers in cytological or histological samples is a valid method [21].

The study of driver mutations in the primary tumor and in the pleural tumor focus suggests that different alterations are present between these two sites. This discordance in the mutational spectra in tumor cells is a phenomenon that adds to the complexity of the pathogenesis of MPE. Pleural effusion is associated with a very broad spectrum of mutations. EGFR oncogene mutations are more common in cases of pleural metastases or those with pleural fluid compared to primary tumors (26% vs. 15%, respectively) [20]. The frequency of KRAS mutations in cases of MPE were lower than in patients with lung cancer at 19% versus 33%, respectively [20].

Other molecular alterations such as ROS1 or ALK have also been described as more frequent in patients with MPD, but especially in the young population [22,23].

When we carried out this study, the recommendations for patients with MPE due to NSCLC were to test at least three biomarkers for targeted therapies, including EGFR, ALK, and ROS1, and with the possibility of treatment within clinical trials for tumors with other molecular alterations, with the aim of greater individualization of treatment and therapeutic efficacy and with considerable progress in identifying new mutations in patients with advanced NSCLC [5,6].

In particular, the development of specific antibodies against the programmed death (PD-1) receptor, programmed death-ligand 1 (PD-L1), and the cytotoxic T-lymphocyte-associated protein 4 receptor in the therapeutic strategy of NSCLC either in first- or in second-line settings have led to unprecedented, prolonged survival for a proportion of these patients [12]. However, recent studies show that MPE by NSCLC is associated with worse survival, even if the PD-L1 expression is greater than 1% [24]. In our series, 35% of the patients had a PDL1 expression greater than 1% of tumor cells, this being the most predominant, but nevertheless, the survival rate is similar to other series without the expression of this marker.

Not having performed the sequencing of mutations in pleural fluid and given the disparity of cell blocks present in the pleural cavity, this could have led to a difference in frequency of the molecular alterations described in our series compared to other studies [21,23].

In relation to survival, MPE patients had a worse survival. Rodriguez EF. [25] conducted a study with the objective of reporting molecular features of pulmonary adenocarcinoma patients presenting with a MPE at the first diagnosis. The control group consisted of patients with a MPE that developed during disease progression. Alterations in EGFR were the most frequent identifiable molecular changes.

Regarding therapeutic options to control pleural effusion, definitive treatment was offered in only 44% of cases. The latest clinical practice guidelines recommend early treatment to control MPE [26,27], which could mean better ECOG and probably greater survival. In our study, 59.4% (31/52) patients were treated with talc pleurodesis (20 patients with slurry pleurodesis and 11 with poudrage pleurodesis). But pleurodesis aims to ease palliative symptoms rather than extend survival.

Concerning the expression of PDL1 in our series, it was found in 31% of patients with DPM. Similar percentages are described in other series [13]. No correlation was found between PDL1 expression and EGFR mutation. The positive expression rate of PD-L1 was not correlated with age and gender in the study population [13].

Our study presents different complications. First, this is a single-center retrospective study. Second, we only included those patients who underwent molecular analysis, although it is true that in our center this is performed in all cases of patients with advanced NSCLC. In conclusion, no significant association was observed between any biomarker and the presence of MPE. This could be due to the small sample size. Therefore, it is thought that environmental factors, in addition to ethnic/racial origin could be related to the disease phenotype, and they could be responsible for the differences between our results and others.

On the other hand, one of the strengths of this study is that it is the first study in a Caucasian population that describes the molecular alterations present in patients with MPE. Future studies could compare their findings with ours.

In conclusion, the EGFR gene mutation was the most frequent molecular alteration found and was associated with a tendency to present metastasis in the form of pleural involvement. Larger-scale cohort studies are necessary to confirm these results and improve knowledge about the molecular alterations in MPE due to NSCLC.

## Figures and Tables

**Figure 1 medicina-60-01804-f001:**
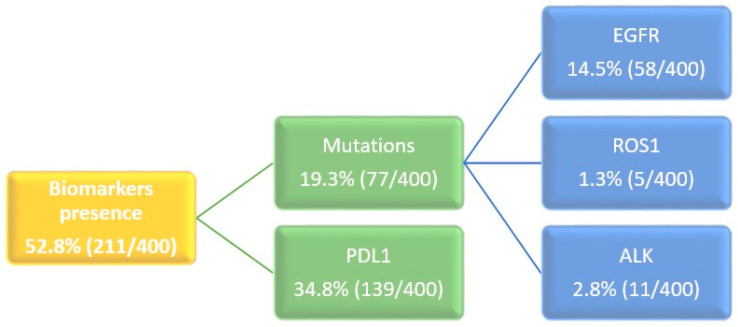
Biomarkers expression.

**Table 1 medicina-60-01804-t001:** Demographic and clinical characteristics of patient included in the analysis.

Variable	N = 400
Age	67.3 ± 10.7 years old
Gender (men/women)	282/118 (70.5%/29,5%)
Smoking	303 (75.8%), Smoking packs year of 41.3 ± 24.7
Risk factors	19.3% (77)
Symtoms - Dyspnea - Chest pain - Haemoptysis - Cough - Constitutional symtoms - Neurological sumptoms	86.5% (346) - 21.2 (73). Average of 2.19 ± 0.7 mMRC scale - 15.4% (53) - 9.9% (34) - 10.4% (36) - 30.4% (105) - 10.8% (43)
Charlson	0.74 ± 1.1
ECOG ≤1 ≥2	1.19 ± 0.9 63.8% (255) 36.3% (145)

**Table 2 medicina-60-01804-t002:** Clinical and demographic differences in patients with and without EGFR presence.

	EGFR Present (N = 59)	EGFR Non-Present (N = 341)	*p* Value
Age	70.4 ± 12 years old	66.8 ± 10.4 years old	0.02
Gender-women	35 (59.3%)	83 (24.3%)	0.0001
Smoking	21 (35.6%)	282 (82.7%)	0.0001
Pleural thikness	8 (13.6%)	17 (5%)	0.2
Pleural nodules	8 (13.6%)	21 (6.2%)	0.05
LENT risk-high	2/14 (14.3%)	19/58 (32.8%)	0.21
Recurrence	29 (51.8%)	98 (35.1%)	0.02
Pleural recurrence (appearance of pleural effusion)	7/29 (24.1%)	16/96 (16.7%)	0.4

Pleural recurrence: appearance of pleural effusion.

**Table 3 medicina-60-01804-t003:** Clinical, demographical and treatment characteristics in patients with or without MPE.

	MPE (N = 116)	Without MPE (N = 284)	*p*
Age (years)	69.9 ± 10.2	66.2 ± 10.2	0.002
Gender- men	71.6% (83)	70.1 %(199)	0.8
Smoking	69% (80)	78.5% (223)	0.05
Symptoms	91.4% (106)	84.5% (240)	0.08
Risk factor	18.1% (21)	19.7% (56)	0.8
ECOG ≤ 1	54.3% (63)	67.6% (192)	0.02
Pleural nodules	13.8% (16)	4.6% (13)	0.002
Pleural thickness	18.1 (28)	1.4 (4%)	0.0001
Presence if mutations and/or PDL1	52.6% (61)	52.8 (150)	1
EGFR	18.1% (21)	13.4% (38)	0.3
Oncological treatment	59.5% (69)	67.6% (192)	0.1

## Data Availability

Dataset available on request from the author.

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
