# Peer review of "Clinical and Molecular Features of Malignant Pleural Effusion in Non-Small Cell Lung Cancer (NSCLC) of a Caucasian Population"

_medicina, 2024, doi:10.3390/medicina60111804_

Round 1

Reviewer 1 Report

Comments and Suggestions for Authors

Thank you for asking me to review this manuscript titled “Clinical and Molecular Features of Malignant Pleural Effusion in Non-Small Cell Lung Cancer (NSCLC) of a Caucasian Population” by Dr. Irene Lojo Rodríguez and her colleagues from the Alvaro Cunqueiro Hospital’s Sanitary Research Institute Galicia Sur (IISGS).

This is a retrospective, single institution study which tried to describe the clinical, pathological and molecular characteristics of Caucasian patients with a malignant pleural effusion (MPE) due to NSCLC and to elucidate factors that can be associated with the presence of biomarkers comparing the results with patients with NSCLC without MPE.

This is a study with a number of methodological and presentation deficiencies that need addressing. Please see my comments below.

Comments:

Ethnicity/racial background was not mentioned in the inclusion criteria.

The inclusion criteria are too broad and include very heterogenous groups!!  For example, all ages where included (even <18 years old), all sexes (how many are there really??), multiple treatment modalities, ect!!

I presume all 400 patients had molecular testing (as implied/covertly stated). Please make this more clear in the inclusion criteria!!

Why where patients who presented with an initial pleural effusion grouped together with patients which developed a MPE later on in treatment?

What do the authors mean by “other causes of effusion were ruled out”? What about atelectasis and reactive pleural effusion due to it?

Why was only overall survival recorded and not recurrence rate and disease free survival?

Make a listing in the text of all the genes (and PDL-1) that where analyzed. Did all patients undergo testing for all genes?

What is Constitutional Syndrome and Asthenia? Please use correct English terminology.

How many patients in total had MPE? Was it as mentioned 116 out of the 400?? This could be a relatively small sample size to do any meaningful statistical analysis!! Was there a power analysis performed to confirm it is adequate?

Was pleural thickness/pleural nodules considered as recurrence? The numbers don’t add up. Was recurrence confirmed histologically? How many patients with thickening or nodules undergo biopsy??

The bottom line is that there was no significant association of any biomarker with the presence of MPE!! This could be a result of a small sample! Please add in the limitations section.

The language of the manuscript needs improvement – better proof reading and editing!

How many variables (and which) were included in the multivariate analysis model? There is a limit on variable number determined by the sample size(minimum 30 patients per variable). Was this rule followed?

Although the aim of the manuscript was supposed to be the association of MPE with gene/PDL1 expression a number of other variables where included/analyzed and eventually presented giving the feeling the authors applied the “drop a line and see what bites approach”!!

The authors mention treatment of MPE but don’t explain/elaborate on what! VATS pleurodesis, talc slurry, indwelling pleural catheter?? The authors need to present these data and include them in the model since they are not all equally efficacious and thus this can be the explanation for the decreased survival!! Please include in the discussion section.

No explanation is given in the discussion why the authors feel their results differ with what is presented in the literature!!

In conclusion, I would like to see a number of corrections made prior to recommending the publication of this work. Thank you and kind regards to all.

Comments on the Quality of English Language

Language editing and proofing is needed!!

Reviewer 2 Report

Comments and Suggestions for Authors

The manuscript presents a detailed analysis of the clinical, pathological, and molecular characterization of patients with malignant pleural effusion (MPE) in the setting of non-small cell lung cancer (NSCLC), focusing on a Caucasian population. This study is relevant given that the diversity of patients and variability in molecular mutations highlight the need for improved characterization of these pathologies, particularly in the case of MPE. The manuscript provides valuable data that may contribute to a better understanding of the molecular profile of patients with MPE and how specific mutations, such as that of the EGFR gene, may influence its development.

Overall, I consider the work meritorious because of the clinical significance of the findings and their potential contribution to the personalized management of NSCLC. However, below, I provide some minor comments that could help improve the clarity and presentation of the results.

Introduction: I recommend enriching this section by including information on lung cancer, its epidemiology, the primary forms of lung cancer, and the most common complications, such as malignant pleural effusion. This will provide the reader with a better frame of reference to understand the study's relevance and the clinical context in which it takes place.

Materials and Methods (Molecular Analysis): Authors should describe in greater detail the methodology used in the molecular analyses, such as Sanger sequencing, in situ hybridization, RT-PCR, and pyrosequencing. For example, it would be useful to include information on the primers used in Sanger sequencing and RT-PCR, which is essential for replicating and understanding the study.

Reviewer 3 Report

Comments and Suggestions for Authors

In the study, the authors determined the “Clinical and Molecular Features of Malignant Pleural Effusion in Non-Small Cell Lung Cancer (NSCLC) of a Caucasian Population”. The topic is relevant, but identified some shortcomings in the manuscript that need to be addressed based on the specific recommendations below:

1.     Lines 14: First define abbreviation MPE, NSCLC.

2.     Lines 31: There is a typo for “survival” and “targeted therapy”.

3.     Improve content of introduction. How MPE is going to affect NSCLC. Discuss it.  DOI: 10.1183/16000617.0019-2016; https://doi.org/10.1186/s12931-024-02684-7;  https://doi.org/10.3389/fonc.2022.961440; https://doi.org/10.1515/biol-2022-0575

4.     Please provide a summary in graphical form including the current incidence and mortality rates of Non-Small Cell Lung Cancer.

5.     The references should be formatted according to the journal style. Add the journal name in italics.

6.     Line 202; 215: Add reference

7.     The quality of the English Language is moderate.

Comments on the Quality of English Language

The quality of the English Language is moderate.

Round 2

Reviewer 1 Report

Comments and Suggestions for Authors

Dear Editor and Authors,

I re-reviewed your edited and resubmitted manuscript titled "CLINICAL AND MOLECULAR FEATURES OF MALIGNANT PLEURAL EFFUSION IN NON-SMALL CELL LUNG CANCER (NSCLC) OF A CAUCASIAN POPULATION" and have taken under consideration your responses to my original commentary.

I am quite satisfied with your revision and most of my concerns have been addressed. The manuscript I feel is quite improved.

I would like prior to final acceptance to ask that:

a) A comment is added in the text in regards to my comment No. 6 regarding reccurence rates and disease free survival.

b) Comments are added in the text in regards to my comments No. 10 regarding reccurence.

Thank you and kind regards.

Author Response

Thanks, we agree with all comments
